# Comparison of the Hydration Characteristics of Ultra-High-Performance and Normal Cementitious Materials

**DOI:** 10.3390/ma13112594

**Published:** 2020-06-06

**Authors:** Haiyun Zhou, Hongbo Zhu, Hongxiang Gou, Zhenghong Yang

**Affiliations:** Key Laboratory of Advanced Civil Engineering Materials, Ministry of Education, Tongji University, Jiading 201804, China; 1830693@tongji.edu.cn (H.Z.); ghx950823@163.com (H.G.); ZSDJ1995@163.com (Z.Y.)

**Keywords:** ultra-high-performance cementitious materials (UHPC), normal cementitious materials (NC), silica fume, hydration

## Abstract

The hydration mechanism of ultra-high-performance cementitious materials (UHPC) departs considerably from that of normal cementitious materials (NC). In this study, the strength, isothermal calorimetry, chemical shrinkage, X-ray diffraction (XRD), and thermogravimetry (TG) methods are used to determine the hydration characteristics of UHPC and NC that contain silica fume (SF). A simple device was modified to test the chemical shrinkage for long-term growth, and the ultimate chemical shrinkage is obtained by semi-empirical formula fitting. It is found that the degree of hydration of UHPC is significantly lower than that of NC. The hydration kinetics analyzed using the Krstulovic-Dabic model shows that the hydration process of NC is type NG-I-D, which is characterized by gentle and prolonged hydration. However, the hydration of UHPC is type NG-D with the distinguishing features of early sufficiency and later stagnation. The growth of the strength, exothermic evolution, and phase development of UHPC is decelerated as the hydration process proceeds, which confirms the weak development tendency of hydration at the later stage. In addition, the effect of SF on the hydration of UHPC is minor, and the higher content of SF is beneficial to the hydration at the later stage.

## 1. Introduction

UHPC is an innovative cementitious material that possesses ultra-high mechanical properties and ultra-high durability and is generally prepared by the tightest accumulation classic theory [1,2,3]. The compounding principle of UHPC makes it very different from traditional concrete in performance, and, in the final analysis, the hydration mechanism of the two systems is notably different. 

Up to now, three methods have been commonly used to study the hydration of cementitious materials [4,5,6], namely, the quantitative X-ray diffraction (QXRD) method, isothermal calorimetry, and chemical shrinkage test method. Several studies have discussed the hydration process of cement by the above methods. Lam [7] and Huang [8] have identified that the composites with low w/b (<0.38) showed a slow exothermic development in the later stage of hydration, which attributes to the insufficient moisture. However, this interesting conclusion is a conjecture of the researchers, which has not been proved by experimental explanation. The Krstulovic-Dabic model with calorimetric data utilized by Yan [9] and Zhang [10] indicated that the hydration process of low w/b composites is significantly different with a high w/b cementitious system. However, the w/b used by authors is all higher than 0.25, and the hydration kinetics of ultra-low w/b composites has not been regarded as a research target. The QXRD and thermogravimetry (TG) methods were used by Korpa [11] to investigate the phase development of NC and UHPC, which displayed that the amount of unhydrated cement in UHPC is considerably higher than that of NC. However, it is difficult to guarantee the accuracy of the QXRD method. The short-term hydration is often evaluated using chemical shrinkage behavior [12], but the long-term hydration characteristics of cement-based materials are still unclear. A theory suggested by Zhang [4] can predict ultimate chemical shrinkage using hydration formula and phase density, however, which did not take into consideration the influence of admixture and w/b on hydration. Additionally, the microscopic analysis conducted by Shen [13] and Ali [14] showed extensive SF agglomeration in the hardened paste of UHPC, and there was no obvious Ca(OH)_2_ crystalline phase even under the conditions of high-temperature excitation or long-term hydration.

Although various studies on the hydration characteristics of cement-based materials have been conducted, systematic studies on the hydration of UHPC is still limited, especially in the case of long-term hydration. Moreover, the difference in properties and hydration behavior between UHPC and NC considered in the previous study are partly speculative. The preparation with ultra-low w/b and a large amount of cement will definitely lead to the hydration characteristics of UHPC significantly different from that of NC. In order to analyze the unique hydration properties of UHPC, including hydration exothermic potential, chemical shrinkage level, and phase development, etc., it is necessary to compare the hydration difference between NC and UHPC systematically. In this study, multiple measurements were performed to study the hydration characteristics of the two systems simultaneously, which is also beneficial to the guarantee of experimental reliability. Moreover, the chemical shrinkage method was modified for testing the long-term volume change of the composites, and a semi-empirical formula was used for predicting chemical shrinkage limit. 

## 2. Materials and Methods 

### 2.1. Raw Materials 

P·O 52.5 Portland cement (C, Jiangnan-Onoda Cement Co., Ltd., Nanjing, China) and silica fume (SF, Qinghe Fengye metal materials Co., Ltd., Xingtai, China) were used as cementitious materials, the components of which are listed in Table 1.

The particle size distributions (PSD) of binders were measured using a laser scattering technique with ethanol dispersion (LS-230), as shown in Figure 1. In the figure, C-V and SF-V mean the differential volume fraction of cement and silica fume, respectively, C-CV and SF-CV mean the cumulative differential volume of cement and silica fume, respectively.

Type PC-200 superplasticizer with a water reduction rate of more than 20%, manufactured by Yingshan New Materials Co., Ltd. (Shanghai, China), was applied to improve the fluidity of the mortar. ISO standard sand and river sand (particle size less than 5 mm) were used as the fine aggregate.

### 2.2. Measurement Methods

#### 2.2.1. Preparation of UHPC

The mixing procedure of UHPC paste described in Figure 2 has excellent fluidity. The mixer of JJ-5A was used for mortar preparation, which is a unified piece of equipment specified in GB/T17671-1999 [15], and the volume of paste is within one third of the mixer volume. The ratios of water/binder, sand/binder, and superplasticizer/binder were fixed as 0.18, 0.8, and 0.6%, respectively. Another five groups of NC mixtures were made in accordance with the GB/T 17671-1999, which present a visible difference from the UHPC. All specimens were placed in a curing room at a temperature of 20 ± 2 °C and relative humidity more than 90%.

The proportion between two cementitious materials is shown in Table 2, where N and U indicate NC and UHPC, respectively, and NSX/USX represents the substitution of cement with 10% or 20% SF. For example, US10 represents the binary binder system, and the mass of cement: SF = 9:1.

#### 2.2.2. Krstulovic-Dabic Model

The hydration of cement based on the Krstulovic-Dabic model [16] is divided into three basic processes, namely, nucleation and crystal growth (NG), interaction at phase boundaries (I), and diffusion (D). For the cement paste, the three processes mentioned above are not necessarily available, and the overall hydration rate depends on the control process with the slowest reaction rate. The Krstulovic-Dabic model is a prominent way to study the hydration kinetics of cementitious materials, the mathematical representations of which are as follows:(1)$$\mathrm{NG}:\;{\left[-\ln\left(1-\alpha \right)\right]}^{\frac{1}{n}}=K_{1}\cdot \left(t-t_{0}\right)={K_{1}}^{\prime }{}^{\;}\cdot \left(t-t_{0}\right)$$
(2)$$I:\;{\left[1-{\left(1-\alpha \right)}^{\frac{1}{3}}\right]}^{1}=K_{2}\cdot r^{-1}\cdot \left(t-t_{0}\right)={K_{2}}^{\prime }{}^{\;}\cdot \left(t-t_{0}\right)$$
(3)$$D:\;{\left[1-{\left(1-\alpha \right)}^{\frac{1}{3}}\right]}^{2}=K_{3}\cdot r^{-1}\cdot \left(t-t_{0}\right)={K_{3}}^{\prime }{}^{\;}\cdot \left(t-t_{0}\right)$$

The differential equations are:(4)$$\mathrm{NG}:\;\frac{d\alpha }{\mathrm{dt}}=F_{1}\left(\alpha \right)={K_{1}}^{\prime }{}^{\;}\cdot n\cdot \left(1-\alpha \right)\cdot {\left[-\ln\left(1-\alpha \right)\right]}^{\frac{n-1}{n}}$$
(5)$$I:\;\frac{d\alpha }{\mathrm{dt}}=F_{2}\left(\alpha \right)={K_{2}}^{\prime }{}^{\;}\cdot 3{\left(1-\alpha \right)}^{\frac{2}{3}}$$
(6)$$D:\;\frac{d\alpha }{\mathrm{dt}}=F_{3}\left(\alpha \right)={K_{3}}^{\prime }{}^{\;}\cdot 3\frac{{\left(1-\alpha \right)}^{\frac{2}{3}}}{2-2{\left(1-\alpha \right)}^{\frac{1}{3}}}$$
where α is the degree of hydration; K_1_(K_1_^’^), K_2_(K_2_^’^), and K_3_(K_3_^’^) are the reaction rate constants of the three hydration processes, respectively; n is the reaction order; t_0_ indicates the end time of induction period; and r is the diameter of the particles involved in the reaction.

The main factor that controls the hydration reaction is different in different gradations [17]. Generally, the moisture supplied for the reaction is enough at the onset of the hydration, and NG plays a leading role. As the reaction proceeds, the hydration products created continuously cover the surface of the unhydrated cement particles, so the migration of ions is blocked. Then, the hydration must continue at the boundary or by diffusion, denoting that the reaction is controlled by I or D in this stage.

#### 2.2.3. Chemical Shrinkage Test

ASTM C1608 [18] specifications are only suitable for measuring chemical shrinkage at an early age (less than 24 h), because the deformation of the rubber stopper leads to insufficient air tightness during long-term testing. Accordingly, a simple device (Figure 3) was made using rigid connection instead of rubber stopper to test chemical shrinkage for 60 d, and the experimental steps are as follows:(1)Firstly, put the weighed material powders, including the water reducer, into a pycnometer, and add water. A small electric whisk was used to mix the powder and water into a uniform slurry.(2)Secondly, vibrate the pycnometer slightly to remove air bubbles, and pour oil into the bottle until its neck.(3)Then, insert the graduated tube into the pycnometer and seal it with paraffin. Pour oil from the top of the graduated tube to raise the liquid level until it reaches the scale line.(4)Finally, place the pycnometer in a water bath at 20 °C, and record the descent height of the liquid level at specific intervals.

The record intervals increased with the increase in hydration time. It was recorded every 0.5 h for the first three hours, followed by recording every 1 h until 6 h, and subsequently, the record times were selected as 12 h, 24 h, 48 h, 72 h, 7 d, 14 d, 28 d, and 60 d. Additionally, the operation takes a certain time, so the zero of time is approximately 10 min after mixing.

#### 2.2.4. TG and XRD

Thermogravimetry (TG, TG 209 F3 Tarsus^®^, Bavaria Asia, Germany) was conducted for thermal analysis at a heating rate of 20 °C min^−1^ from 20 to 1000 °C in a nitrogen atmosphere. The specimens without sand was prepared, broken, dried at 40 °C for 48 h, and separated by a 75 µm sieve for a microstructural test. The equipment of D/MAX 2550VB3 + /PC manufactured in Japan (Kumamoto) was adopted for XRD characterization at a scanning rate of 5°/min from 5° to 90°.

## 3. Results and Discussion

### 3.1. Mechanical Property

Testing in accordance with GB/T 17671-1999 was performed to determine the strength of the cubes of 40 × 40 × 160 mm^3^, and the test age was 3, 7, and 28 d.

As illustrated in Figure 4, the strength of UHPC is significantly higher than that of NC, and the compressive strength of the former at 28 d is approximately twice that of the latter. The excellent mechanical properties of UHPC may be attributed to the compact structure, which is facilitated by the high binder content and low w/b [19]. Nevertheless, the ratio of the 3-d-strength vs. 28-d-strength of NC is 0.7, while that of UHPC is 0.8. The result suggests that the strength development of UHPC is stagnant in the later stage, and the conclusion is also in agreement with the analyses reported in the references [20,21].

The effect of SF on the strength of UHPC is similar with NC. In general, the compressive strength first increases and then decreases with the increase in SF dosage, which is generally coincident with the change rule of flexural strength. 

SF is thought to have the ball effect, filling effect, and pozzolanic effect, which can promote the mechanical properties of cement by optimizing the microstructure [22]. So, the compressive strengths of NS10 and US10 increase by 2.62% and 4.72%, respectively, compared to that of control samples (N and U). On the other hand, the fluidity of the paste reduces as a result of the higher SF fraction, deriving from the negative effects of superplasticizer absorbed and water wrapped by SF. According to Powers’ Theory [23], the bubbles caused by the low flowability are responsible for the decrease in strength. This means that the strength reduces unless the working performance is firstly met [24].

It must be identified that the purpose of the paper is investigating differences between UHPC and NC matrix, so the compressive strength of UHPC is lower than 150 MPa, which was prepared without adding steel fiber or raising under high temperature.

### 3.2. Analysis of Calorimetric Data

The rate of heat release during the hydration of NC and UHPC was monitored by the calorimeter for three days, whose measurement is compliant with ASTM C1679-17 [25]. The curves in Figure 5 show the results.

As shown in Figure 5, the induction period of NC is completed during the first 1.5–2 h after the cement is mixed with water, while that of UHPC is 3–5 h. SF has similar effects on the induction period of NC and UHPC, that is, the induction period is advanced when SF is added. Meanwhile, the second hydration heat release peak of the UHPC is sensitive to the SF content, contrary to the NC case. For UHPC, the peak height shows a trend of first increasing, and then decreasing with the increase in the SF content. Overall, the hydration heat release rate of the U-series is lower than that of N-series, and the influence of admixture content on the rate of the two systems is discordant. This phenomenon can be attributed to the fact that the hydration moisture provided by UHPC is singularly lower than NC, which leads to a high dosage of admixture dissolve difficultly. Coupled with the agglomeration of SF itself and the encapsulation of water, the low heat emission rate of UHPC with 20% SF is observed. 

The hydration heat evolution can be evaluated by the area under the hydration heat emission rate-time envelope, which is depicted in Figure 6.

Figure 6 shows that the hydration heat release of UHPC is notably lower than that of NC for the curing age of three days, namely, the Q_3d_ of N-series is 252–300 J/g, while that of U-series is in the range of 166–182 J/g. For UHPC, the difference in Q_3d_ between the SF-containing UHPC and pure-mix U is within 6%, which indicates that SF has little effect on the early heat release and hydration of UHPC. The same conclusion was also given by Lothenbach [26] and Weerdt [27]. Furthermore, during the first 18 h of hydration, the SF plays a role in controlling the internal temperature of the NC and UHPC, which has been proven by other researchers as well [10,28]. However, the effect is not maintained over time, and varies with the SF content and cementitious mixes.

The ultimate heat emission can be estimated by Knudsen’s extrapolation formula:(7)$$\frac{1}{Q}=\frac{1}{Q_{\max}}+\frac{t_{50}}{Q_{\max}\left(t-t_{0}\right)}$$
where Q is the hydration heat emission at time of t (h); Q_max_ is ultimate hydration heat emission value; t_50_ is the half time of hydration reaction; and t_0_ is the end time of induction period.

The main factors of the hydration heat emission are estimated in Table 3. In addition, the fitting results obtained by Knudsen’s extrapolation formula are summarized in Figure 7.

Considering Table 3, the calculated Q_max_ of NC is 31.25–52.94% higher than that of UHPC. From the age of three days to the time of complete hydration, the heat emission of NC increases by 20.02–42.55%, while the increase for UHPC is only about 15%. Furthermore, the effect of SF content on the increase of hydration heat emission (Q_3d_ to Q_max_) of NC and UHPC is consistent, that is, the increment for 20% SF cement systems is superior to that of 10% SF cement systems. NS20 and US20 increases by 42.55% and 17.89%, respectively, while NS10 and US10 increases by 20.02% and 14.81%, respectively. It can be inferred that the degree of hydration of the samples with 20% SF incorporated is higher in the later stage.

According to the previous studies [29,30,31,32,33], the mechanism of SF action on cement hydration is two-fold. On the one hand, SF facilitates hydration: (1) the pozzolanic reaction between SF and hydration products Ca(OH)_2_ creates C-S-H, which promotes the initial hydration of the cement. However, some scholars hold that the SF slows down the further hydration of cement by increasing the density of C-S-H around the cement particles and hindering the diffusion; (2) the large surface of SF provides a nucleation site and C-S-H precipitation space for hydration. On the other hand, SF hinders the hydration: (1) SF is coated on the surface of cement particles to restrain the contact between cement and water; (2) the water provided for hydration is encapsulated because of the SF clumping with high surface energy; (3) SF adsorbs a certain quantity of superplasticizer molecules, thereby weakening the damage by the agent to the flocculation structure of cement. In summary, the effect of SF on cement hydration is complex, and the dominant mechanism is different for different stages and systems. According to the results of the experiment, the SF affects the hydration of the high w/b system considerably, while it has little effect on the ultra-low w/b cementitious materials that are without enough moisture and hydration capacity.

The Krstulovic-Dabic model in ‘Section 2.2.2’ was used to describe the hydration process, and the reaction rates F_1_(α), F_2_(α), and F_3_(α) of the three hydration processes were calculated based on the hydration heat data of the samples. The results are plotted in Figure 8.

As shown in Figure 8, curves F_1_(α), F_2_(α), and F_3_(α) can satisfactorily simulate the hydration rate curve obtained by experiments in sections, which indicates that the hydration of cement-based materials is a complex process involving multiple mechanisms. The two transition points, from process NG to process I and from process I to process D, are labeled as *α*_1_ and *α*_2_, respectively. The thermogram shows that the hydration process of sample N is of type NG-I-D, the hydration process of which is gentle and prolonged. However, that of sample U is of type NG-D, which may be attributed to the hydration characteristics of UHPC. For cementitious materials with ultra-low w/b, the water provided for hydration is sufficient at relatively early ages, but the quantity of products raises rapidly, which is in opposition with the diffusion process without enough water for ions migration. Therefore, the hydration is directly from NG to D, which has the slowest reaction rate. The hydration characteristics of NC and UHPC reflected by the thermograms coincides with the conclusions obtained by the strength test and calorimetric analysis, which reveals that the growth of the strength and exothermic evolution of UHPC is lower than that of NC. Moreover, the t_50_ of NC is significantly longer, which conforms to the feature of prolonged hydration. 

### 3.3. Prediction of Ultimate Chemical Shrinkage 

The evolution of the chemical shrinkage is plotted as a function of the age in Figure 9.

Figure 9 shows that the chemical shrinkage of U-series is lower than that of N-series, and the chemical shrinkage is enhanced by the higher addition of water, which has been verified by Geiker and Knudsen [34]. The SF has a converse effect on the chemical shrinkage values of NC and UHPC; that is, the chemical shrinkage of NC is enhanced in the presence of SF, while the shrinkage of UHPC reduces instead. In addition, the cement-based materials with incorporated SF shows a higher growth rate than the pure paste, especially in group NS20 and US20, which is similar to the development of the hydration heat emission. 

As highlighted by Mounanga [35] and Zhang [4], the degree of hydration can be reflected by the chemical shrinkage of cement to a certain extent. Hence, UHPC with ultra-low w/b develops a lower hydration degree than NC for a given age. In the meantime, it is difficult for SF to promote hydration in UHPC mixes because of its flocculation, and the water provided for hydration is insufficient. It is also worth noting in Figure 9 that the chemical shrinkage of each group reaches the plateau at the age of 100 h, after which the increase in contraction is minor. Based on this, the early chemical shrinkage rate is studied to analyze the hydration characteristics of the cementitious materials as below. 

The chemical shrinkage rate vs. age (from 0 h to 24 h) are plotted in Figure 10.

From Figure 10, the curves of the chemical shrinkage rate can be divided into three segments: (1) the deceleration period—the magnitude of chemical shrinkage rate is the highest when cement comes into contact with water, but the value decreases as the clinker phase C_3_A is consumed quickly; (2) the acceleration period—the dashed box on Figure 10 highlights that the chemical shrinkage rate of NC has an ascending portion at 5–12 h (the acceleration period should start before 5 h due to the limitation of test operation), which is related to the second hydration heat release peak in Figure 5. The acceleration period of UHPC is not pronounced during the first 24 h of hydration; (3) the plateau, the hydration reaction is gradually controlled by diffusion with the increase in the quantity of products after 12 h of curing, and, so, the chemical shrinkage rate tends to be stable. Figure 10 shows that the chemical shrinkage rate of UHPC is higher than that of NC at early stage (0–12 h), which is then exceeded by NC in the acceleration period. The relationship between the hydration degree and w/b has been proved in other studies. Mounanga [36] deems that “the lower is the w/c ratio, the faster the saturation degree of hydrates is reached”; accordingly, the cementitious paste with low w/b shows a superior hydration degree during the first few hours of hydration. However, with the gradual consumption of water, the system with high w/b shows up in the hydration rate.

A semi-empirical formula for predicting the ultimate chemical shrinkage (CS_U_) of cement-based materials was proposed by Xiao [37], which originates from the growth curve model.
(8)$$\mathrm{CS}\left(t\right)=\frac{{\mathrm{CS}}_{U}\cdot t^{a}}{t^{a}+b}$$
where CS(t) is the chemical shrinkage value as a function of the hydration age; t is the time of the hydration; a is a constant, which is related to the factors that affect hydration rate; b is the time to reach half CS_U_. If t_0.5_ is the time of 0.5CS_U_, b = $t_{0.5}^{a}$.

Equation (8) is used for fitting the experimental data of the chemical shrinkage, as shown in Figure 11, and the calculated CS_U_ values are listed in Table 4.

Figure 11 reveals that the fitting curves agree well with the actual shrinkage, and the semi-empirical formula is also applicable to UHPC with an extremely low water-binder ratio. The formula takes into account the effect of the mix proportion on the shrinkage of the hardened paste, but it is not suitable for fitting the chemical shrinkage data from the short duration test, as the calculated CS_U_ values tend to be lower than the true value. Additionally, the CS_U_ of UHPC is lower than that of NC, and the deterioration of CS_U_ is exceptional when SF is used in the UHPC paste.

Bouasker [38] has signified that the relationship between chemical shrinkage and the degree of hydration is quasi-linear, which can be expressed as $\alpha \left(t\right)=\frac{\mathrm{CS}\left(t\right)}{{\mathrm{CS}}_{U}}=\frac{Q\left(t\right)}{Q_{\max}}$. In order to verify the relationship, the curves of hydration degree reckoned by chemical shrinkage and hydration heat release, respectively, are depicted in Figure 12. 

As shown in Figure 12, for NC, the hydration degrees calculated by chemical shrinkage and exothermic evolution, respectively, show the same trend with the temporal variation, which suggests that there is indeed a quasi-linear relationship between shrinkage and hydration for ordinary concrete with high w/b. Despite the similar development trend, for UHPC, there is still a big difference between the two hydration degrees. In the early stage of hydration, CSα is sensitive to the SF, but not to Qα, which is reversed in the later stage of hydration. It is inferred that the degree of hydration can be reflected by chemical shrinkage, but the relationship between them is not completely quasilinear.

### 3.4. Calculation of Phase Content from Combined TG /XRD

TG and DTG curves are shown in Figure 13, which expresses the hydration process clearly.

As shown in Figure 13, three obvious weight loss peaks are observed [39]. Firstly, the dehydration peak of C-S-H and ettringite (AFt) at 40–230 °C, including the loss of evaporated adsorbed water at 40–105 °C, the loss of C-S-H interlayer water at 80–210 °C and the loss of AFt crystal water at 150–210 °C. Secondly, the dehydration peak of Ca(OH)_2_ (CH) at 400–500 °C, and no major thermal events have been observed in this temperature range, which facilitates the determination of CH content. Finally, the decarbonation of calcium carbonate in the vicinity of 720 °C, and Ca(CO)_3_ originates from the reaction of CH with CO_2_, which needs to be taken into account in the calculation of CH production. 

Typical plots of DTG curves for all samples collected at 3 and 28 d are given in Figure 14.

Figure 14 indicates that the dehydration peak at 28 d is sharper than that at 3 d, which demonstrates the increase in C-S-H and AFt with the increase in elapsed age. There is no considerable difference in the appearance of the CH-dehydration region at the two ages, indicating that CH content has not increased significantly over time. Furthermore, the two former peaks of UHPC move to the low temperature zone in contrast with NC, which illustrates the hydration product content of UHPC is less than that of NC for a certain age.

The CH content can be calculated by the formula proposed by Zhu [40], namely: (9)$$\mathrm{CH}=74\left[\frac{{\mathrm{TG}}_{1}}{18}+\frac{2}{3}\cdot \frac{{\mathrm{TG}}_{2}}{44}\right]$$
where TG_1_ and TG_2_ indicate the CH-dehydration rate and Ca(CO)_3_-decarbonation rate, respectively.

Variation of CH content with hydration age is presented in Figure 15.

For pure cement samples N and U, the CH content increases with the increase in curing time, resulting from the improvement of the hydration degree of cement. However, for cement that contains SF, the CH content reduces with time, and the major decrease is induced by the higher level of SF addition, because the enhanced pozzolanic effect consumes a lot of CH. The observations previously made regarding the hydration rate also apply to CH formation, that is, a higher content of SF is beneficial to the hydration at the later age. At the age of 28 d, the CH content of UHPC (4.20–7.48%) is considerably lower than that of NC (6.31–15.92%). Coupled with the lower hydration heat release and chemical shrinkage, suggesting that the hydration degree of UHPC is lower than that of NC at the same curing age.

The XRD patterns of all mixes at an age of 3 and 28 d are plotted in Figure 16.

The hydration degree of cementitious materials can be characterized not only by the CH content but also by the residual amount of the reactants in the hardened paste. Figure 16 shows that the influence of w/b and admixture on the peak intensity of CH is consistent with the influence on CH content, as measured by TG. There is a direct proportionality between the mass fraction of the crystalline phase and the area of the strongest peak in the XRD spectrum. Based on this principle, the proportion of C_3_S in UHPC is higher than that of NC, which also reveals the low hydration degree of UHPC. Moreover, for sample NC, the C_3_S content decreases significantly with the curing age from 3 to 28 d, while the C_3_S content of UHPC is almost unchanged with time, which can be inferred from the shape of the peaks. The phenomenon indicates, again, that the hydration process of UHPC plateaus off in the later stage. 

## 4. Conclusions

The compressive strength of UHPC is more than twice that of NC at a given age, and the strength development of UHPC shows a weak tendency. An optimal addition of SF is beneficial to the strength of cement-based materials, otherwise, the strength of hardened paste will be obviously reduced due to the gain in pores.

The induction period of UHPC ends several hours later than that of NC, and the predicted Q_max_ of NC is 31.25–52.94% higher than that of UHPC. The hydration process of NC is type NG-I-D, while that of UHPC is type NG-D, which may be attributed to insufficient water for diffusion. The t_50_ of NC is significantly longer than that of UHPC, which conforms to the hydration features of two different systems, that is, the hydration of NC is gentle and prolonged, while the hydration of UHPC shows tardy progress over time. Moreover, the cementitious materials with a higher SF content present a higher hydration degree in the later stage.

The chemical shrinkage of UHPC is lower than that of NC, and the effects of SF on the shrinkage of NC and UHPC are converse, which inhibits the contraction of UHPC. In addition, UHPC with ultra-low w/b show superior hydration during the first few hours of hydration. However, with the gradual consumption of water, the system with high w/b shows a higher hydration rate.

The semi-empirical model is favorable for predicting the ultimate chemical shrinkage of UHPC, but the premise is not to use the short-term shrinkage data. The relationship between the chemical shrinkage and hydration degree is not completely quasilinear.

For SF-containing cement, the consumption of CH by the pozzolanic reaction reduces the CH content in hardened mixes. The CH content of UHPC is 4.20–7.48%, which is significantly lower than that of NC with 6.31–15.92%. This indicates that the hydration degree of UHPC is lower than that of NC for a given age. The C_3_S content of NC decreases significantly with the curing age from 3 to 28 d, while that of UHPC is almost unchanged with time, which can be interpreted as the hydration evolution of UHPC being slow in the later stage.

## Figures and Tables

**Figure 1 materials-13-02594-f001:**
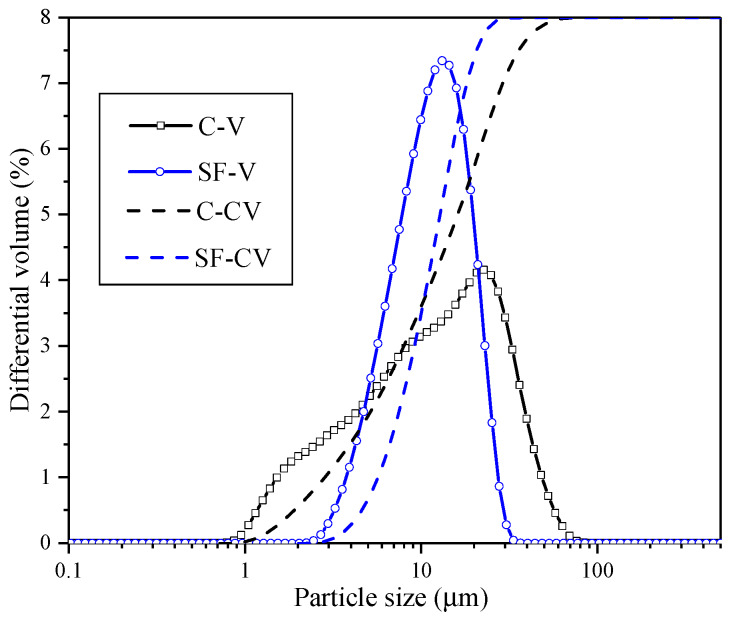
Particle size distributions PSD of cementitious materials.

**Figure 2 materials-13-02594-f002:**
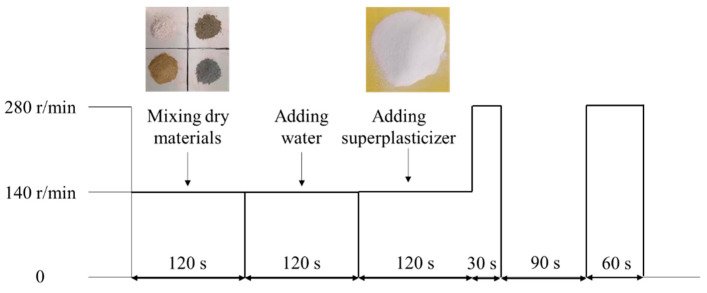
Preparation process of ultra-high-performance cementitious materials (UHPC).

**Figure 3 materials-13-02594-f003:**
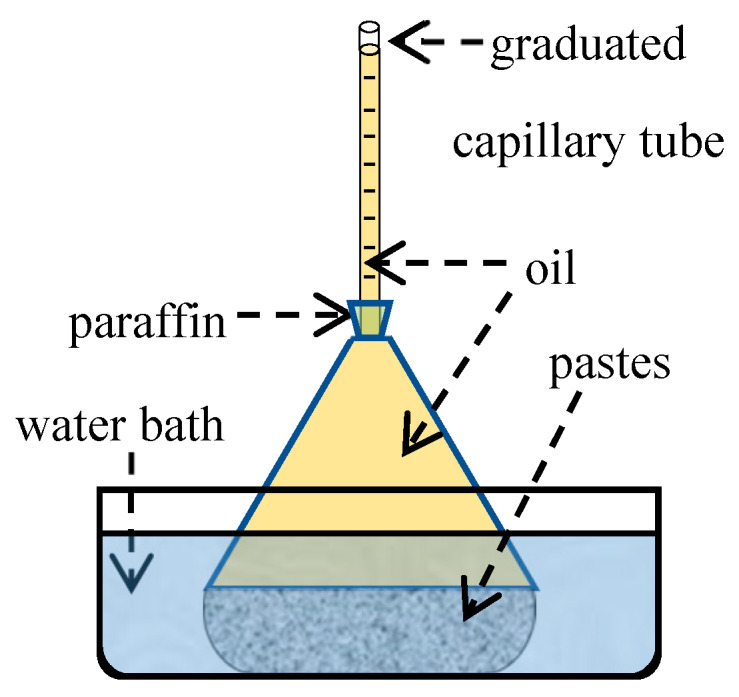
Experimental setup for measuring chemical shrinkage.

**Figure 4 materials-13-02594-f004:**
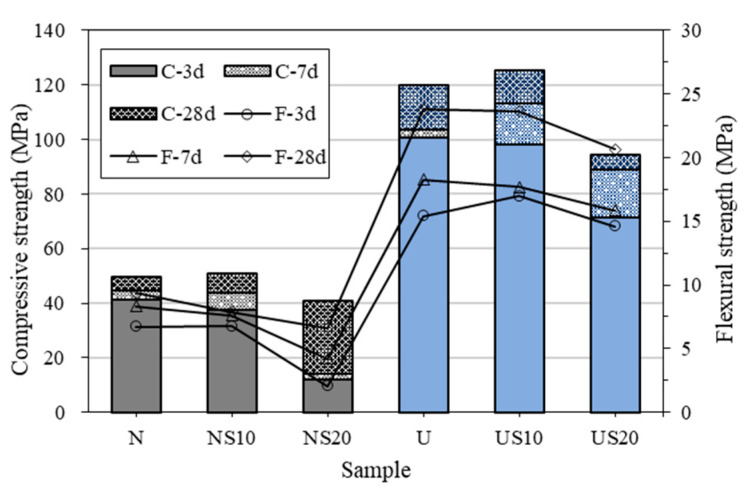
Effect of mix proportion on strength of samples.

**Figure 5 materials-13-02594-f005:**
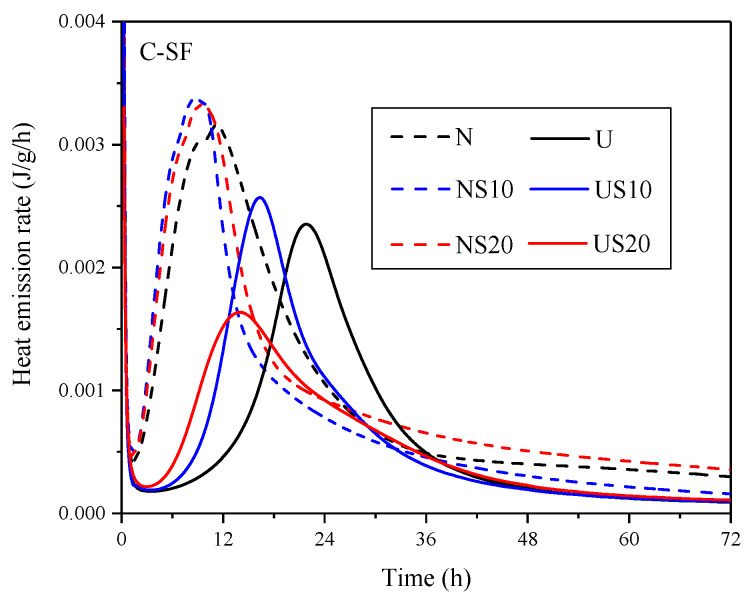
Curves of the heat release rate during hydration.

**Figure 6 materials-13-02594-f006:**
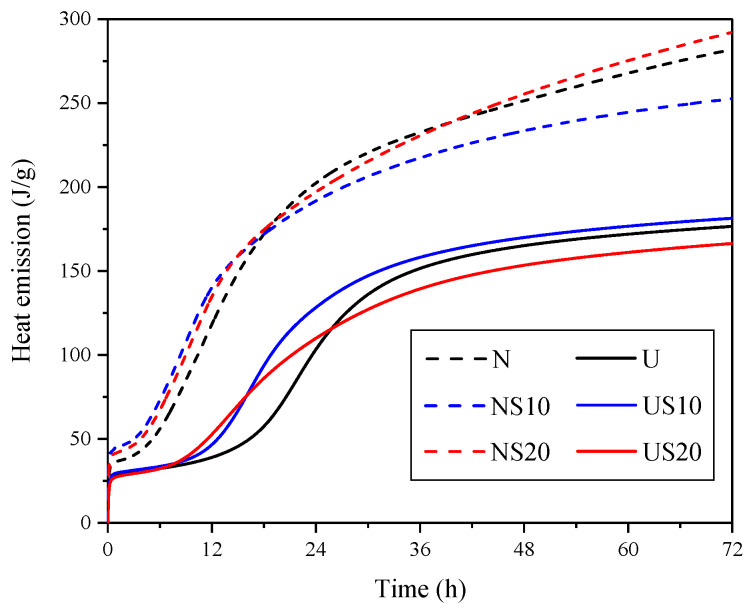
Curves of hydration heat release.

**Figure 7 materials-13-02594-f007:**
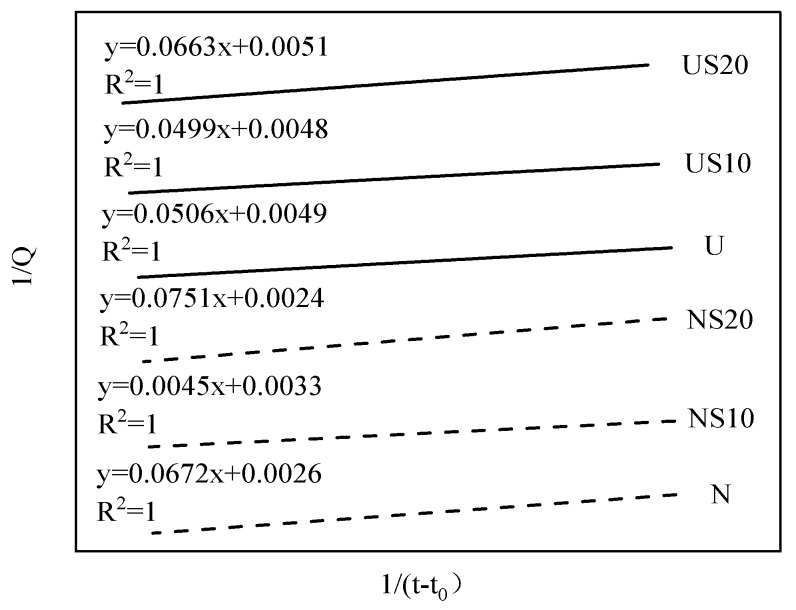
Fitting process by Knudsen’s extrapolation formula.

**Figure 8 materials-13-02594-f008:**
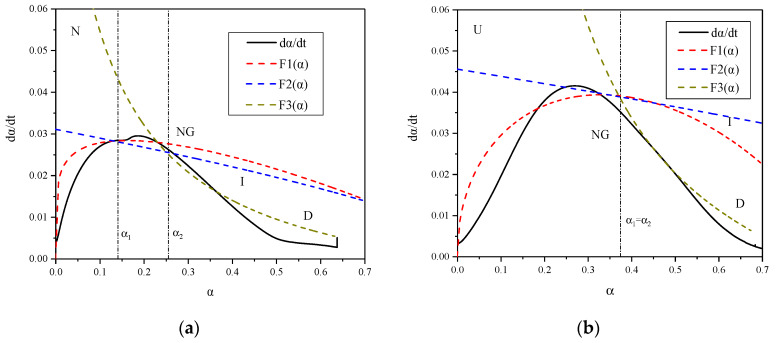
Thermograms of neat paste (**a**) N and (**b**) U.

**Figure 9 materials-13-02594-f009:**
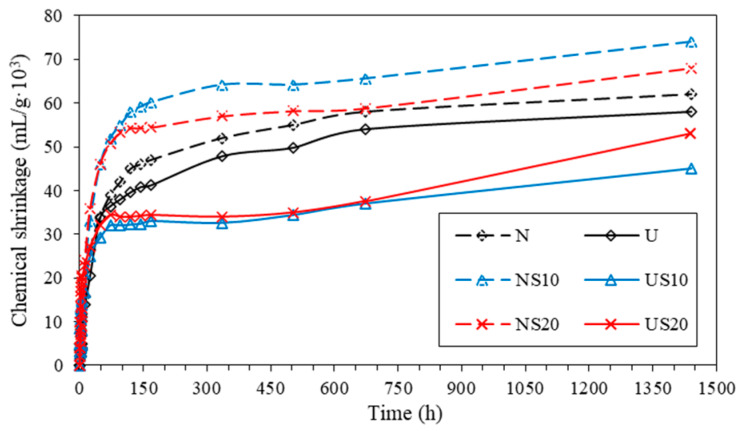
Chemical shrinkage for 60 d.

**Figure 10 materials-13-02594-f010:**
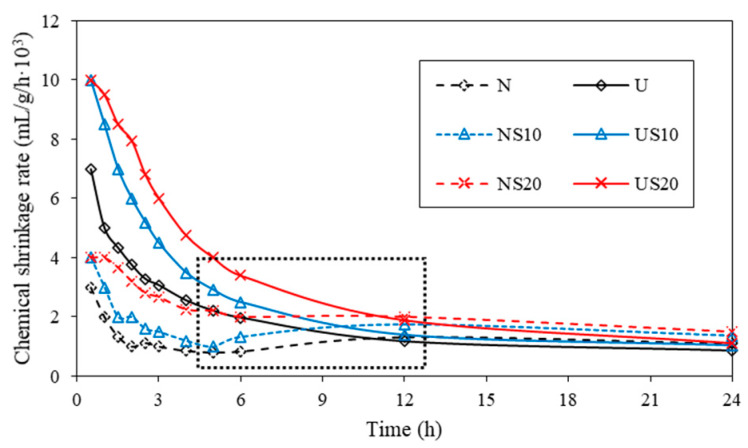
Chemical shrinkage rate at early age (0–24 h).

**Figure 11 materials-13-02594-f011:**
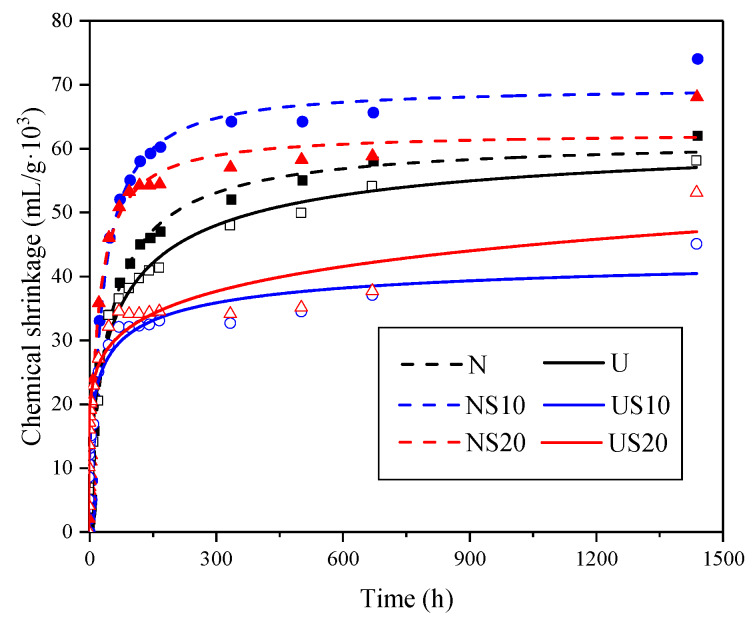
Fitting results obtained by the semi-empirical formula.

**Figure 12 materials-13-02594-f012:**
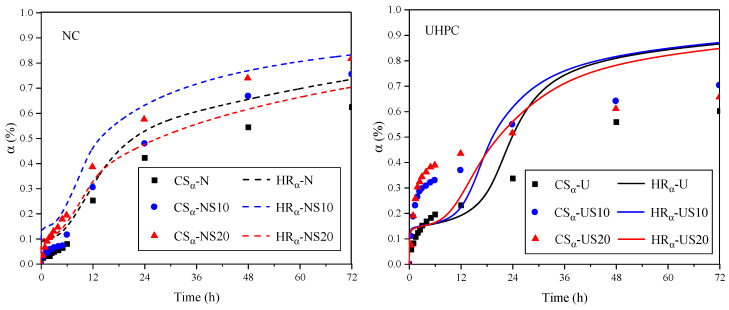
Relationship between hydration degree, chemical shrinkage, and hydration heat evolution. (CSα-chemical shrinkage; Qα-hydration exothermic amount).

**Figure 13 materials-13-02594-f013:**
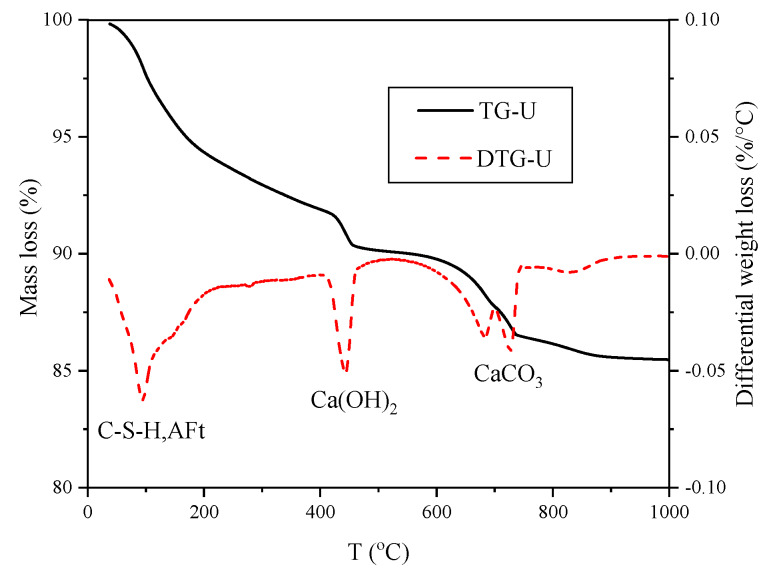
TG and DTG curves of U for 28 d.

**Figure 14 materials-13-02594-f014:**
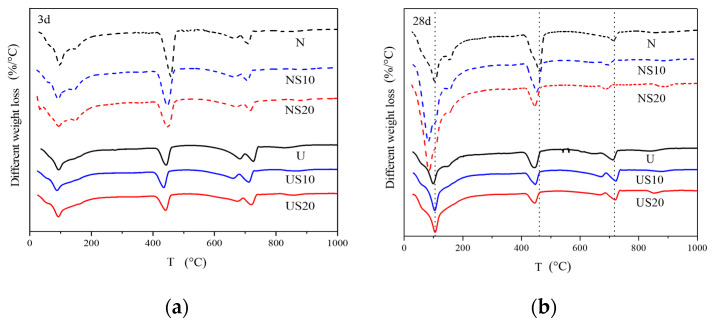
DTG curves for (**a**) 3 d and (**b**) 28 d.

**Figure 15 materials-13-02594-f015:**
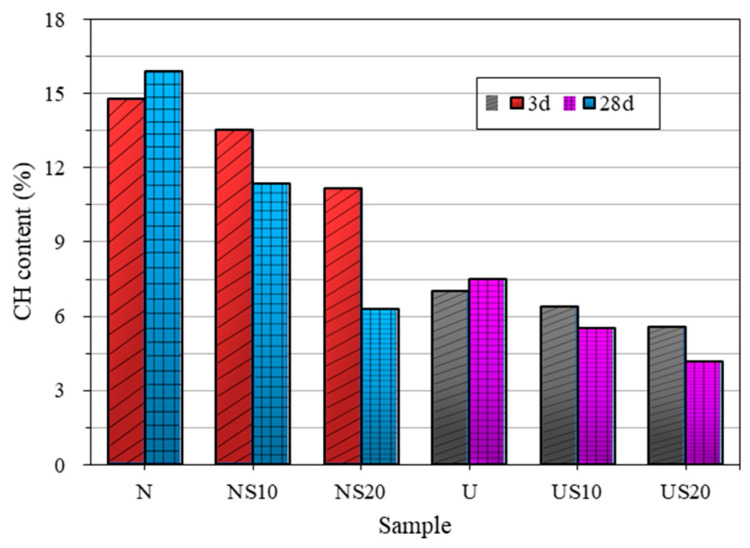
Ca(OH)_2_ (CH) content for 3 d and 28 d.

**Figure 16 materials-13-02594-f016:**
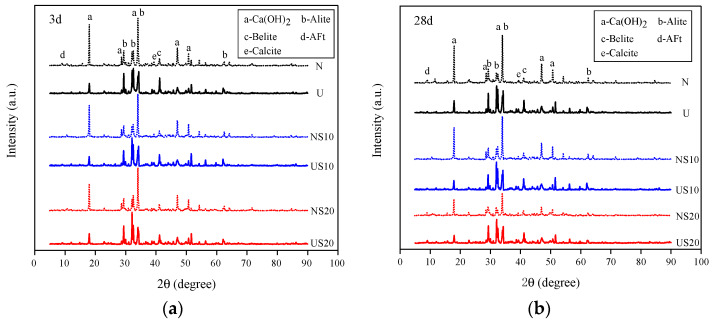
X-ray diffraction (XRD) patterns at (**a**) 3 d and (**b**) 28 d.

**Table 1 materials-13-02594-t001:** Chemical composition of cementitious materials (wt.%).

Materials	CaO	SiO_2_	Al_2_O_3_	Fe_2_O_3_	SO_3_	MgO	K_2_O	Na_2_O
C	65.00	20.90	4.56	3.23	2.65	0.87	0.71	0.10
SF	0.46	97.77	0.22	0.07	0.16	0.44	0.42	0.15

**Table 2 materials-13-02594-t002:** Designed sample composition.

NC	Cement	SF	UHPC	Cement	SF
N	1	-	U	1	-
NS10	0.9	0.1	US10	0.9	0.1
NS20	0.8	0.2	US20	0.8	0.2

**Table 3 materials-13-02594-t003:** Main parameters of hydration heat evolution.

Sample	Q_3d_ (J/g)	t_0_ (h)	t_50_ (h)	Q_max_ (J/g)	Knudsen’s Extrapolation Formula
N	282.17	1.33	25.85	384.62	y = 0.0672x + 0.0026
NS10	252.47	1.47	13.64	303.03	y = 0.0045x + 0.0033
NS20	292.29	1.24	31.29	416.67	y = 0.0751x + 0.0024
U	176.83	3.53	10.33	204.08	y = 0.0506x + 0.0049
US10	181.45	3.28	10.40	208.33	y = 0.0499x + 0.0048
US20	166.33	3.02	13.00	196.08	y = 0.0663x + 0.0051

**Table 4 materials-13-02594-t004:** Main parameters of the semi-empirical formula.

Sample	CS_U_ (mL/g·10^3^)	t_0.5_ (h)	R^2^
N	62.46 ± 1.47	40.49	1.00
NS10	68.87 ± 1.21	27.81	0.99
NS20	62.35 ± 1.32	19.37	0.99
U	60.46 ± 2.51	55.95	1.00
US10	45.57 ± 3.97	20.83	0.97
US20	52.29 ± 0.00	22.51	1.00

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
