# Peer review of "Comparison of the Hydration Characteristics of Ultra-High-Performance and Normal Cementitious Materials"

_materials, 2020, doi:10.3390/ma13112594_

Round 1

Reviewer 1 Report

The paper entitled “Comparison of hydration characteristics of ultra-high-performance and normal cementitious materials” experimentally compared the hydration properties of ultra-high-performance cementitious materials (UHPC) and normal cementitious materials (NC).

In general, the methods used in this study and the research presented is interesting and systematic however the changes are required in the introduction part where the author needs to elaborate the sentence written in line 55-57 (Therefore, it is important to investigate the unique hydration properties of UHPC by comparing the difference between NC and 56 UHPC systematically). Considering the results obtained in this study, I wonder what unique hydration properties were investigated? This line needs to be elaborated.

Regarding to the method used to determine chemical shrinkage (In section 2.2.3). Author referred the method ASTM C1608, in which taking up to 24 hours reading is suitable, but to take reading for 60 days, author modified the method. I would like to know how accurate is this method to take the long term readings? And what modification makes this method accurate to take long term reading? Would you elaborate it for readers’ convenience?

Author Response

    On behalf of my co-authors, we appreciate Materials for giving an opportunity to revise our manuscript. We thank for your careful reading and constructive comments. Those comments are all valuable for improving our paper, and also important for guiding our researches. We have carefully taken the suggestions into the consideration which highlighted in yellow. 

Reviewer 2 Report

This study describes the experimental determination of hydration characteristics of Ultra high performance cementitious and normal cementitious mixtures with 0, 10 and 20 % cement replacement by silica fume. 

This paper is well written overall with only a few English language and style errors. The following suggestions can be considered to improve the manuscript:

In text, instead of using "~" which denotes "about", please use "-" to denote that a value is between two numbers. 

In all figures, please check that units of measurement appear on axes in this form, e.g. Strength (MPa) instead of Strength/MPa, for consistency. "/" should only be used as "per", to avoid confusion.

In Table 1, please declare whether the compositions listed are in % form.

Section 2.2.1 should include information regarding the type of mixer used, and speed vlaues (low and high) to prepare the UHPC paste, as well as the volume of paste prepared.

Please revise the x-axis label in Figure 14.

Author Response

   On behalf of my co-authors, we appreciate Materials for giving an opportunity to revise our manuscript. We are very honored to have your recognition of this paper. In addition, we thank for your careful reading and constructive comments. Those comments are all valuable for improving our paper, and also important for guiding our researches. We have carefully taken the suggestions into the consideration which highlighted in yellow. 

Reviewer 3 Report

In this article the Authors presents a study on the hydration mechanism of an ultra-high-performance cementitious material. Portland cement, silica fume and PC-200 superplasticizer were used as materials, and a series of characterizations were carried out to evaluate the hydration of the proposed cementitious material in comparison with a normal one. The hydration kinetics, by Krstulovic-Dabic Model, showed that the hydration process of the normal cement is based on two transition points, characterized by gentle and prolonged hydration, while the hydration process ultra-high-performance cement showed one transition point, characterized by early sufficiency and later stagnation, due to insufficient water for the migration of ions, causing a slower rate.

In my opinion, the topic is very interesting and falls within the scopes of the Materials, however the drafting of the manuscript should be improved. The  objective of the article is clearly developed and the research design is adequate, moreover the characterization techniques are sufficient to satisfy the scope of the work. Some suggestions follow:

  1. The English should be improved.
  2. The punctuation should be revisited.
  3. The bibliographic references should not be superscript in the text.
  4. The Authors should specify the method used for quantitative XRD.
  5. In general, the introduction should be improved, in particular from line 34 to 44. Rather than a list a discussion should be developed, trying to argue.
  6. The equation are low quality images, the equation editor should be used.
  7. Line 216 “bythe”

Author Response

(The authors gave the same response as above.)
